# Eco-Friendly Bioemulsifier Production by *Mucor circinelloides* UCP0001 Isolated from Mangrove Sediments Using Renewable Substrates for Environmental Applications

**DOI:** 10.3390/biom10030365

**Published:** 2020-02-27

**Authors:** Nathália S. A. A. Marques, Israel G. Sales da Silva, Davi L. Cavalcanti, Patrícia C. S. V. Maia, Vanessa P. Santos, Rosileide F. S. Andrade, Galba M. Campos-Takaki

**Affiliations:** 1Northeast Biotechnology Network, Federal Rural University of Pernambuco, Recife 52171-900, Pernambuco, Brazil; nathaliasa13@hotmail.com (N.S.A.A.M.); israelgsds@gmail.com (I.G.S.d.S.); patriciacvsm@gmail.com (P.C.S.V.M.); nessapimentel4519@hotmail.com (V.P.S.); 2Doctorate Program in Biological Sciences, Federal University of Pernambuco, Recife 50870-420, Pernambuco, Brazil; davicavalcanti25@gmail.com; 3Nucleus of Research in Environmental Sciences and Biotechnology, Catholic University of Pernambuco, Recife 50050-590, Pernambuco, Brazil; rosileide_fontenele@yahoo.com.br

**Keywords:** natural emulsifier, agro-industrial waste, toxicity, emulsifying activity, anionic emulsifier.

## Abstract

The successful production of a biosurfactant is dependent on the development of processes using low cost raw materials. In the present work, an economically attractive medium composed of corn steep liquor and waste cooking oil was formulated to maximize the production of bioemulsifier by *Mucor circinelloides* UCP0001. A central rotational composite design was applied to statistical validation of the production. The emulsifying properties, stability under extreme conditions, its toxicity character, and the characterization of the bioemulsifier were determined. The best condition for biomolecule synthesis occurred in the assay 2 containing 4% of corn steep liquor and 3% waste soybean oil and exhibited 100% emulsification index for canola oil and petroleum, as well as excellent emulsifying activity for canola oil and burned engine oil. The nutritional factors studied showed statistical relevance, since all linear, quadratic effects and their interactions were significant. The bioemulsifier showed 2.69 g/L yield and the chemical character of the molecule structure was identified by FT-IR (Fourier Transform Infrared) spectroscopy. The bioemulsifier showed no toxicity to *Artemia salina* and *Chlorella vulgaris*. Stable emulsions were obtained under extreme conditions of temperature, pH, and salinity. These findings contribute to understanding of the relationship between production, physical properties, chemical composition, and stability of bioemulsifier for their potential applications in biotechnology, such as bioremediation of hydrocarbon-contaminated soil and water.

## 1. Introduction

Recently, there has been growing interest from the food and cosmetics industries for green raw materials. Environmental concern and increased consumer awareness of the use of natural ingredients as substitutes for synthetic additives have resulted in numerous studies on screening biological sources (such as plants and microorganisms) for surfactants and emulsifiers of natural origin [1].

Surfactants are surface active molecules that adsorb the oil–water (emulsion) and air–water (foam) interfaces and help in the formation and stabilization of emulsions and foams [2]. Surfactants, on the other hand, are low molecular weight molecules being amphiphilic in nature, with a hydrophilic group and a hydrophobic tail. Its surface activity derives from a balance between hydrophobic and hydrophilic portions in the molecule [1,2].

The two types of molecule can be distinguished based on their structure and physicochemical properties. Chemically, bioemulsifiers are complex mixtures of heteropolysaccharides, lipopolysaccharides, lipoproteins, and proteins [3]. The combination of polysaccharide, fatty acid, and protein components gives bioemulsifiers better emulsifying potential and emulsion stabilization capacity [2,3].

They are able to reduce surface tension to form emulsion droplets and foam bubbles as well as form an adsorbed layer of steric stabilization at the interface (long chain polymers adsorbed onto the surface of the particle) [1]. Conventional production of microbial bioemulsifier is still costly, one of the factors contributing most to the high cost is the use of synthetic nutritional sources. A promising strategy to make the cost of the process economically viable is the insertion of renewable nutritional sources from agro-industrial waste and by-products. In this sense, several low-cost alternative substrates such as frying residual soybean oil and corn steep liquor are explored as substitutes for synthetic carbon and nitrogen sources [4]. A tool combined with biotechnological processes is the use of experimental designs to optimize the production of bioemulsifiers and biosurfactants [5]. The most requested statistical models are the response surface methodology (RSM) and the factorial design. For optimization, researchers generally evaluate different aspects such as crop conditions and nutritional factors, often including carbon and nitrogen concentrations and their proportions [6].

The global biosurfactant/bioemulsifier market is expected to reach $2.2 billion by 2019, based on a growth rate of 3.5% per year. In addition, the global projected production market is estimated at 476,512.2 tons due to increased demand from Asia, Africa, and Latin America, which account for 21% of total production [6].

The aim of this work was to investigate the microbial transformation of a Brazilian agro-industrial fatty waste from soybean oil and byproducts of corn steep liquor as alternative nutritional sources for the economically attractive production and optimization of the *Mucor circinelloides* UCP0001 bioemulsifier, and evaluating the toxicity using different bioindicators *Artemia salina* and *Chlorella vulgaris*.

## 2. Materials and Methods

### 2.1. Microorganisms

*Mucor circinelloides* UCP0001 was isolated from mangrove sediment from Rio Formoso municipality, Pernambuco-PE, Brazil. The strain was deposited with the Bank of Cultures UCP, correspond to “Universidade Católica de Pernambuco”, which has been registered in the World Federation Cuture for Collection-WFCC. The strains were maintained at 5 °C in the Synthetic Medium for Mucorales—SMA. The culture was transferred to Potato-Dextrose-Agar (BDA) medium, pH 5.5, where the components were liquefied and autoclaved at 121 °C for 20 min. The fungus was maintained in BDA medium at 5 °C and monthly was transferred. *Chlorella vulgaris* microalgae was collected from Lagoa do Carro city, Pernambuco, Brazil, and was deposited in the Nucleus of Research in Environmental Sciences and Biotechnology (NPCIAMB).

### 2.2. Substrates

The substrates used for the production of the bioemulsifier were corn steep liquor, kindly provided by Corn Products Brazil Ingredients (Cabo-PE) and the post-frying soybean oil was obtained from informal commerce.

### 2.3. Experimental Planning

Response surface methodology (RSM) was used for experimental design of the rotational central composite design (RCCD) type to optimize bioemulsifier production. Thus, a RCCD 2^2^ with four axial points and four central points was applied to analyze the influence of nutritional parameters (carbon source and nitrogen source) on bioemulsifier production. The nutritional parameters mentioned were determined as independent variables (frying residual oil and corn steep liquor) and the emulsification activity as the response variable. Independent variables and their experimental ranges are presented in Table 1. Experimental data were analyzed using the response surface and contour curve of Statistica^®^ version 10 software (Statsoft Inc, Tulsa, OK, USA) and adjusted for a polynomial model.

### 2.4. Bioemulsifier Production

The inoculum of *M. circinelloides* UCP0001 was performed by transferring spores grown in BDA medium (potato, dextrose, and agar) to distilled water, which resulted in a spore solution of 10^7^ sporangiospores/mL. Aliquots (1 mL) of the spore suspension were inoculated into the production medium containing NH_4_NO_3_ (0.1%), KH2PO4 (0.02%), MgSO4.7H2O (0.02%), basal medium proposed by Luna [7] and different concentrations of corn steep liquor and residual frying oil according to the RCCD 2^2^ factorial design (Table 1). The submerged culture was incubated for 96 h at 28 °C, pH 5.0 with 150 rpm orbital shaking. The medium was autoclaved at 121 °C for 20 min.

### 2.5. Emulsification Index

To determine the emulsification index values, the samples were centrifuged at 10,000 rpm for 15 min. A total of 1 mL of hydrophobic substrates (canola oil, corn oil, motor oil, and crude oil) and 1 mL of cell free metabolic fluid were added to test tubes. Subsequently, the tubes were vortexed for 2 min and allowed to stand for 24 h to measure the emulsion index. The percentage of emulsification index (E_24_) is given as the percentage of the height of the emulsion layer divided by the total height of the liquid column [8]. Assays were performed in triplicate.

### 2.6. Emulsification Activity

For determination of emulsification activity, the metabolic fluid was centrifuged at 4000 rpm for 20 min. Then 2 mL of cell-free supernatant, 2 mL of 0.1 M sodium acetate buffer, and 1 mL of hydrophobic substrate (burnt motor oil and canola oil) were added to graduated tubes. The mixture was vortexed for 2 min. After 10 min of rest, the emulsions were collected with the aid of a Pasteur pipette, placed in a cuvette, and finally read on a spectrophotometer at a wavelength of 540 nm [9]. Samples were performed in triplicate.

### 2.7. Analysis of Emulsion Drops 

To investigate the size and influence of droplets on emulsions formed by the bioemusifier, the droplet diameter was measured using an optical microscope (Olympus BX50, California, USA).

### 2.8. Bioemulsifier Stability

Initially the study of bioemulsifier stability was performed using 20 mL of cell-free metabolic liquid for heat treatment at 4, 60, and 121 °C for 1 h and cooled to room temperature for subsequent measurement of emulsification index (EI_24_). To evaluate the efficiency of the biomolecule at different pH, its pH was adjusted to 4, 7, and 9, then the emulsification index was measured. Finally, the samples were tested at the following NaCl concentrations: 2%, 15%, and 30% [10].

### 2.9. Bioemulsifier Isolation

The bioemulsifier production was isolated in the best condition from cell free metabolic liquid was precipitated by ethanol (1:2 *v/v*), as described by Bueno et al. [11]. The precipitate was allowed to stand for 24 h at 4 °C, then centrifuged at 5000 rpm at 15 min and evaporated to remove residual ethanol, then the bioemulsifier was lyophilized and its yield calculated by gravimetry expressed in g/L. The crude sample was then dissolved in distilled water and dialyzed using a membrane with a diameter of 2.5 cm and a pore size of 10 kDa. The procedure was performed for 72 h changing the water three times a day.

### 2.10. Toxicity of Bioemulsifier Using Micro-Crustaceans as Indicators

The lethality test against *Artemia salina* is an important method used in studies with natural products to evaluate the toxic potential of extracts and isolated substances. For hatching of the larvae, a salt solution in distilled water containing sea salt (30 g/L) was prepared, the pH adjusted to 8.0, and maintained at 28 °C under 100 W light for a period of 48 h [12]. Then 10 nauplii were collected and transferred to penicillin vials containing 0.1%, 0.5%, and 1% of isolated bioemulsifier, crude bioemulsifier, and chemical surfactant (Triton X100), the latter used as a positive control. After 24 h, they were analyzed to record the number of live larvae. *A saline* solution without the sample was used as negative control. The number of live larvae in relation to the increased concentration of the samples was used to calculate the mortality rate values. The experiment was performed in triplicate.

### 2.11. Bioemulsifier Toxicity Using Microalgae 

Different species of microalgae respond differently to dissolved, potentially toxic compounds. Some species are more sensitive than others, the most appropriate way to select the species is to take into account their ecological importance or abundance in local freshwater or marine waters. In this sense, *Chlorella vulgaris* was used as a test organism for toxicity determination [13]. The microalgae was cultured in BG-11 liquid medium at pH 7 under white light and 12-12 h photoperiod (light/dark), room temperature 25 °C, constant aeration for 168 h. Aliquots (10 mL) of 10^6^ cells/mL in the exponential phase were inoculated into vials containing 0.1%, 0.5%, and 1% of the bioemulsifier. Chemical surfactant (Triton X100) was tested at the same concentrations for positive control and BG-11 liquid medium was used as negative control [14].

### 2.12. Biochemical Nature

The total carbohydrate content of the bioemulsifier was determined by the phenol sulfuric acid method according to the methodology described by Dubois et al. (1956). Total lipids were quantified using chloroform and methanol in different proportions (2:1, 1:1, and 1:2, *v/v*), the organic extract was evaporated by rotary evaporator, and the lipid content determined by gravimetry [15]. Protein quantification was performed using the Labtest Diagnostica S.A. (Brazil) kit for colorimetric determination of total proteins by Biuret reagent.

### 2.13. Characterization and Ionic Character

The functional groups of the bioemulsifier were identified by the infrared spectroscopy. Infrared spectra were recorded on a Mattson 1000 FT-England FT-IR spectrometer within the wavelength range of 500–4000 cm^−1^. The ionic charge of the bioemulsifier was determined by measuring the zeta potential, 100 V, 1/4 scale, at a temperature of 22 °C.

## 3. Results 

### 3.1. A New Bioemulsifier Production

In this sense, alternative substrates were determined according to the RCCD 2^2^ factorial design. The experimental design allowed us to investigate the influence of concentrations and their interactions of agro-industrial substrates on bioemulsifier production, validated by the emulsification activity response variable (Table 2).

It is noted that the recovered *M. circinelloides* supernatant presented emulsification capacity above 50% for all hydrophobic substrates, both plant and petrochemical origin. It was found that the highest emulsification indexes (EI_24_) as well as the highest emulsification activities (EA) were present in test 2 (3% waste cooking oil and 4% corn steep liquor). Canola oil and petroleum were 100% emulsified, both emulsions were considered stable since the criterion established for emulsion stabilization is the ability of a bioemulsifier to preserve at least 50% of the original emulsion volume 24 h after its formation (Figure 1A). Figure 1B shows the optical microscopy analysis of the emulsion formed between the burnt engine oil and the *M. circinelloides* metabolite. Analysis revealed the presence of large globular-looking droplets in the emulsion with burnt motor oil. The larger droplets had a size between 3 and 4 µm, however, average particles were observed measuring between 2 and 2.5 µm, as well as particles smaller than 1 µm. Additionally, the formation of the physical stability of the emulsion provided by the aggregation of dispersed particles is remarkable in the microscopic image, however, the absence of coalescence phenomenon was observed. In addition, few spaces between the droplets were found (Figure 1B).

### 3.2. Statistical Validation Maximization of Bioemulsifier Production 

The response surface method (RSM) was used to construct a model for modeling the emulsification activity of canola oil and engine burnt oil (Figure 2 and Figure 3). The following quadratic polynomial equations showed better data fit for the maximum values of EA for canola oil and burnt engine oil, respectively:Z = 1.65 − 0.05(X) − 0.16(X^2^) + 0.001(Y) − 0.11(Y^2^) + 0.18(XY) + 0.9(1)
Z = 2.64 − 0.05(X) − 0.17(X^2^) + 0.16(Y) − 0.05(Y^2^) + 0.13(XY)(2)

In the response surface analysis (RSM) for canola oil EA, it were observed that in the central values of the determined levels of waste cooking oil and corn steep liquor were the best results for the condition EA. On the other hand, analysis of the RSM graph for EA in engine burnt oil indicates that the central values of the corn steep liquor concentration and higher values of waste cooking oil of the culture medium influenced the highest values of EA. In addition, by detecting that the quadratic effects for the two independent variables are considerable, this indicates that the optimal levels are within the range of the experimental region. Similar results were observed for the emulsification of canola oil, the high significant values of the quadratic effect reinforced that the optimal levels were observed in the central points.

The analysis of variance (ANOVA) shows that the explained variance value (coefficient of determination—R^2^) of the predictive model was 98.32 (Table 3), which indicates that the proposed mathematical model explains in 98.32% the reproducibility of the data, response variable, emulsification activity of canola oil.

On the other hand, the ANOVA for the engine burned oil EA indicated that the displayed data show that the explained variance value was 65% (Table 4), proposing that the studied mathematical model could explain only 65% of the data variability. Moreover, it is observed that all factors and levels of this study had significant influences on the response variable, including the lack of adjustment demonstrates that the mathematical model is not well adjusted to the data.

In the residual normality the residuals show statistical normality for the reset variable EA in both canola oil and engine burnt oil (Appendix A
Appendix A).

Through the interpretation of the Pareto graphic (Figure 4) it was found that in increasing order of influence on the response variable we have the quadratic effect of the corn steep liquor (Q); the interaction of linear effects of corn steep liquor with frying residual oil (1L by 2L); the quadratic effect of residual frying oil (Q); linear effect of corn steep liquor (2L), have a statistically significant influence. The corresponding values established by Pearson coefficient line (*p* = 0.05), it was also identified that the linear effect of residual soybean oil has no significant influence on the EA response variable for canola oil. It is noticeable that the only factor that has a significant influence on the increase of EA is the linear interaction between the corn steep liquor and the residual frying oil, since the other factors have a significant negative influence, for instance, contribute to the decrease of EA in canola oil.

Analyses for the response variable EA in engine burnt oil are shown in the Pareto graphic (Figure 5). All the effects and their studied standard interactions shown in the figure have a significant influence on the EA response variable in engine oil. In order: linear effect of residual frying oil (2L); quadratic effect of corn steep liquor (Q); linear interaction of the corn steep liquor with the frying residual oil (1L by 2L); linear effect of corn steep liquor; quadratic effect of residual frying oil (Q). The only effects that positively influenced EA in burnt engine oil were the linear effect of the residual frying oil and the linear interaction of the factors. The other factors were significant, however, for the decrease of EA.

### 3.3. Effect of Environmental Factors on Bioemulsifier Stability

Most of the emulsions formed in both types of oils have been shown to remain stable in the different treatments with acid, neutral, and alkaline pH throughout the storage period (Table 5). Canola oil exhibited a slight loss of stability when treated at pH 4 and pH 9. 

Saline treatments performed during the 120 days showed that the lower the NaCl concentration, the higher the stability of both emulsions. However, at higher NaCl levels, the emulsification index for both oils remained above the considered minimum value (50%) (Table 6).

*M. circinelloides* bioemulsifier was able to emulsify almost 100% motor oil for the three types of heat treatments. In addition, for canola oil it obtained excellent emulsification rates of up to 81% in heat treatments of 4 and 60 °C (Table 7). 

### 3.4. Bioemulsifier Toxicity Using Artemia Salina and Chlorella Vulgaris as Bioindicators

The isolated bioemulsifier was showed the minimum dosage (0.1%) against lethality to *A. salina*, obtaining 100% survival. The increase of the bioemulsifier to 0.5% the nauplii were found to survive 80%, and in the maximum sample amount (1%) the nauplii showed 70% survival rate. On the other hand, in the positive control using the chemical surfactant (Triton X100) at the same concentrations mentioned above, only at its minimum dosage was a survival rate of only 30% found. In addition, the chemical surfactant tested at concentrations of 0.5% and 1% resulted in 100% lethality of the test organisms (Appendix A).

Acute toxicity tests were performed with *C. vulgaris* grown in assays containing different concentrations of bioemulsifier and chemical surfactant (triton X100), used as a positive control (Appendix A). At the end of 168 h incubation in the negative control (BG-11 liquid medium) total of 5.26 × 10^4^ cell/mL was found. In the assay containing the minimum concentration (0.1%) of bioemulsifier, a total of 5.03 × 10^4^ cell/mL was observed, in contrast, when the sample dose was increased to 0.5% and 1%, the number of cells was maintained at 4.93 × 10^4^ cell/mL at both concentrations. From these data the absence of acute toxicity of the biomeulsifier, showing no significant difference in the number of cells found in the control and no eutrophication effects, was verified. On the other hand, testing at the same concentrations with Triton X100 chemical surfactant yielded the following cell numbers: 1.74 × 10^4^ (0.1%), 1.43 × 10^4^ (0.5%), and 1.12 × 10^4^ (1%). Triton X100 present in the culture medium reduced the population of *C. vulgaris* by up to 79%.

### 3.5. Yield and Characterization of Bioemulsifier

The use of ethanol resulted in the formation of a precipitate, it was lyophilized and dialyzed, and a yield of 2.69 g/L was obtained. *M. circinelloides* UCP0001 isolated bioemulsifier is composed by 28% carbohydrates, 14% protein, and 40% lipids, in addition, an anionic character of the bioemulsifier was confirmed by the zeta potential.

In elucidating the molecule, the bioemulsifier was subjected to FT-IR analysis (Appendix A) to identify functional groups. Peak 1102 cm^−1^ of the infrared spectrum proves the presence of C-O-C-linked oligo characteristic of carbohydrates. On the other hand, the peak 1652 cm^−1^ refers to the characteristic amide band of α or β secondary structures of proteins. These results prove the constitution of the functional groups belonging to the hydrophilic region of the bioemulsifier molecule. In addition, peak 3232 cm^−1^-exhibited O-H stretch of the hydroxyl groups in carbohydrates. The confirmation of carbohydrate, protein, and fatty acid components in the isolated bioemulsifier assigns them better emulsifying potential and ability to stabilize emulsions.

## 4. Discussion

Problems related to the production of biomolecules arise mainly due to the use of expensive substrates [16,17]. The substrates used in the present study are economically attractive: corn steep liquor, a cheap alternative to synthetic nitrogen sources, and residual frying oil, compared to fresh vegetable oils used [18,19].

Emulsions are known to be biphasic and thermodynamically unstable systems, so there is a need to incorporate an emulsifying agent [20,21]. The bioemulsifier secreted by *M. circinelloides* decreased the interfacial tension between the inner (water) and outer (oil) phases causing these phases to mix and the emulsion to remain stable.

The bioemulsifier is at the oil–water interface, around the internal phase droplets, as a thin layer of adsorbed film on the surface. The film avoids contact and coalescence of the internal phase, and the more resistant and flexible it is, the higher the emulsion stability [20].

A study proposed by Souza et al. [22], reported a bioemulsifier synthesized by *Candida lipolytica* yeast under the same nutritional conditions (corn steep liquor and residual soybean oil) proposed in this work. The emulsion droplets formed by the yeast bioemulsifier obtained globular appearance due to dispersion and flocculation, which tended to attach to larger volumes.

Emulsification activities (EA) obtained by filamentous fungi are considered to correlate strongly with bioemulsifier production [23]. In this sense, the response surface method (RSM) was used to construct a model. A study conducted with RSM to optimize the production of *Haererehalobacter* sp. bioemulsifier and found that the effect of carbon source with NaCl was significant for the increase of bioemulsifier synthesis, efficient in the emulsification of commercial vegetable oils [24].

Moreover, it is worth mentioning that if there is an increase in the concentration of corn steep liquor in the crop there will be a significant reduction for EA. A study also used the response surface method to optimize the production of a bioemulsifier glycolipid synthesized by the marine bacteria *Planococcus* sp. [19]. A recent study selected several cultivation aspects to investigate bioemulsifier production by *Bacillus subtilis* grown in cassava wastewater, and found that only inoculum size was statistically significant in biomolecule production [25].

A study optimized the production through RSM of a biosurfactant agent synthesized by *Candida lipolytica* grown in 5% animal fat and 2.5% of corn steep liquor, obtaining a value of 95.39% in the analysis of variance (ANOVA) [26]. The literature reports that the optimization of bioemulsifier production is still focused on costly cultivation methods, such as *Bacillus subtilis* lipopeptide, grown on synthetic medium containing glucose and yeast extract explained 90% the variability of data [27].

In general, the studied factors demonstrated statistical relevance, since all linear, quadratic effects and their interactions were significant for the increase of emulsification activity. Using the RCCD, mathematical models were found to describe the functioning of these phenomena, with explained variance values above 95%, only for canola oil EA, indicating an excellent explanation of the variability in response.

Aiming at the application of natural emulsifying agents in aquatic and terrestrial environments with high salinity, pH of agricultural soils, environments that absorb much of the heat, or even soils of cold temperate environments, the study of the stability of these molecules is of essential importance. From this point of view, the functionality of the bioemulsifier was evaluated over a period of up to 120 days.

These results corroborate the bioemulsifier synthesized by *Acinetobacter bouvetii* UAM25 grown in glycerol and residual frying oil [3].

Ionic surfactants are very sensitive to electrolytes, this phenomenon occurs because electrolytes directly affect some functional groups of the molecule, consequently decreasing the emulsification index [28]. These results were superior to studies also within 120 days for stability of emulsions formed with *Candida lipolytica* biosurfactant UCP0988 [29]. The data obtained in this analysis corroborate the thermostable bioemulsifier of *Acinetobacter bouvetii* UAM25 [3] and the bioemulsifier produced by *Bacillus* spp. [30]. This fact can be explained due to heat activation, since emulsions are thermodynamically unstable systems and any temperature increase results in increased droplet collision rates [28]. However, high molecular weight bioemulsifiers decrease this phenomenon due to this characteristic.

Toxicity can be defined as the ability of a substance to have a detrimental effect on a living organism, moreover, it depends on the concentration and properties of the chemical to which the organism is exposed and the exposure time [29,31]. Toxicity testing using the micro-crustacean *A. salina* as a bioindicator was performed as a suitable model organism and widely used in acute toxicity studies, and is very sensitive to surfactants [32,33]. These results corroborate the literature, which confirmed that the filamentous fungus biosurfactant *Cunninghamella echinulata* showed no acute toxicity against *A. salina* [34].

These tests confirm the importance of investing more and more in the production of natural surfactants, since chemical surfactants of petrochemical origin are products that are considerably toxic to the environment and the health of organisms. Green and unicellular freshwater algae such as *C. vulgaris* present in this study are often used in toxicity tests because of their rapid growth and their cultures are easily maintained in the laboratory [35]. The use of algae as a biological indicator is important because, as primary producers, they are at the base of the food chain and any change in the dynamics of their communities can affect higher trophic levels of the ecosystem [36].

The result obtained in relation to the yield of isolated biomolecule corroborates the data found in the literature. These describe yields for bioemulsifiers of microbiological origin between 0.1 and 3.15 g/L [25,30,37,38,39]. The yield of the fungal bioemulsifier of this study was superior to the bacterial of *Lactobacillus plantarum*, obtaining only 0.7 g/L of the isolate [39].

## 5. Conclusions

In the present study, the analysis of nutritional factors for the maximization of bioemulsifier production showed statistical relevance, since all linear, quadratic effects and their interactions were significant for the emulsification activities studied. After application of the RCCD, mathematical models were found, which describe the functioning of these phenomena, with explained variance values above 95% for canola oil emulsification activity, indicating an excellent explanation of the variability in response. The bioemulsifier showed no toxicity in relation to the bioindicators, moreover, it formed stable emulsions for a long time under extreme temperature, salinity, and pH conditions. The stability and non-toxicity of *M. circinelloides* UCP0001 bioemulsifier are decisive factors for its implementation in biotechnological applications in agriculture and bioremediation since this molecule has great potential for green technology, carefully designed by methods as a novel microbial emulsifier likely to instigate new collaborative approaches for protection of the biodiversity.

## Figures and Tables

**Figure 1 biomolecules-10-00365-f001:**
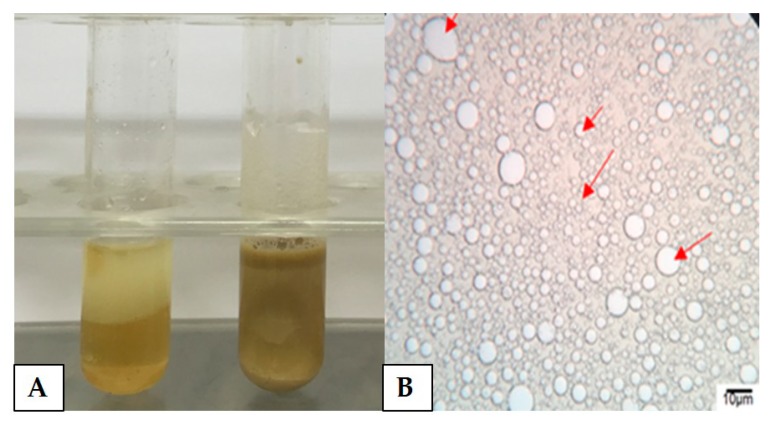
Macroscopic appearance of emulsions formation by *Mucor circinelloides* bioemulsifier using canola oil (near A letter) and burnt engine oil (near B letter), respectively (**A**), and microscopic analysis of the emulsion formed by the bioemulsifier in burnt engine oil (**B**).

**Figure 2 biomolecules-10-00365-f002:**
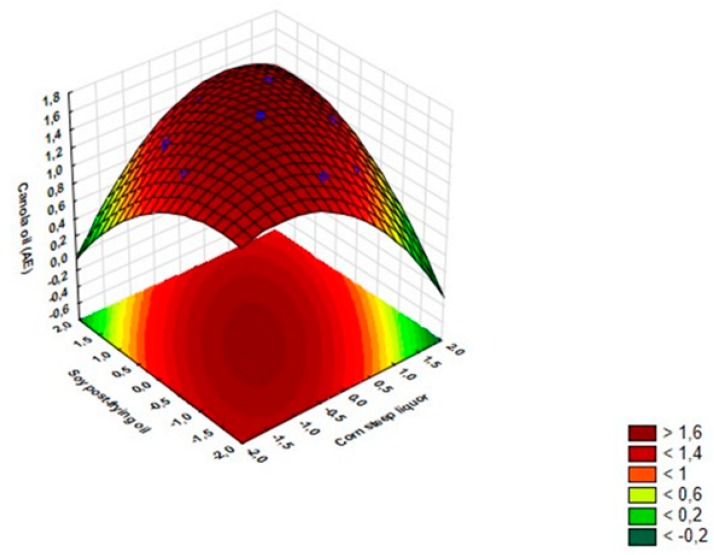
Response surface graphic for interactive effect between concentrations of corn steep liquor (%) and waste cooking oil (%) on emulsification activities (EA) in canola oil.

**Figure 3 biomolecules-10-00365-f003:**
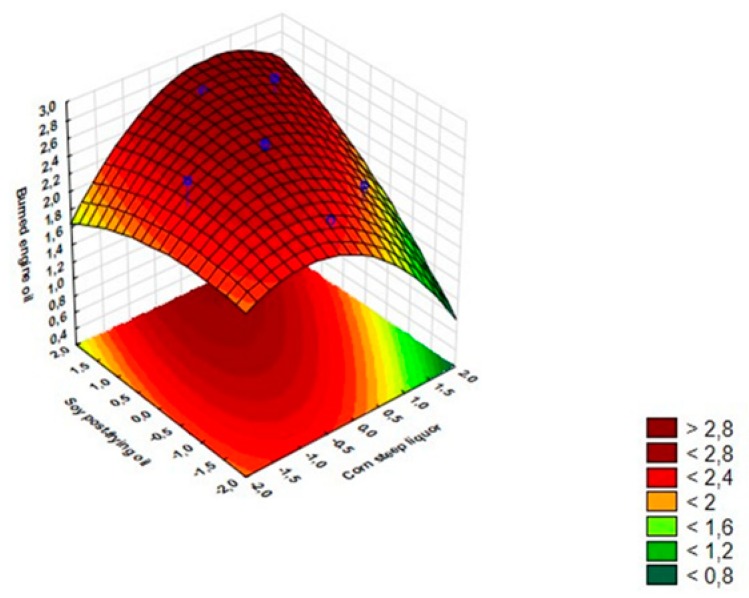
Response surface graph for interactive effect between concentrations of corn steep liquor (%) and waste cooking oil (%) on EA in motor oil.

**Figure 4 biomolecules-10-00365-f004:**
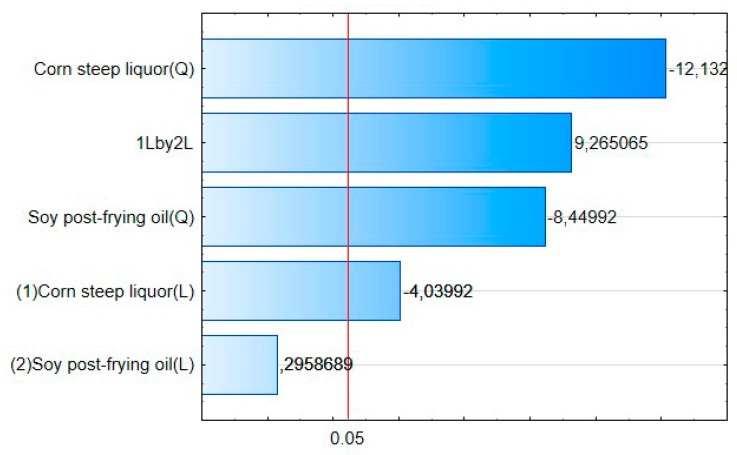
Influence of the independent variables (corn steep liquor and frying residual oil) and their interactions on canola oil EA.

**Figure 5 biomolecules-10-00365-f005:**
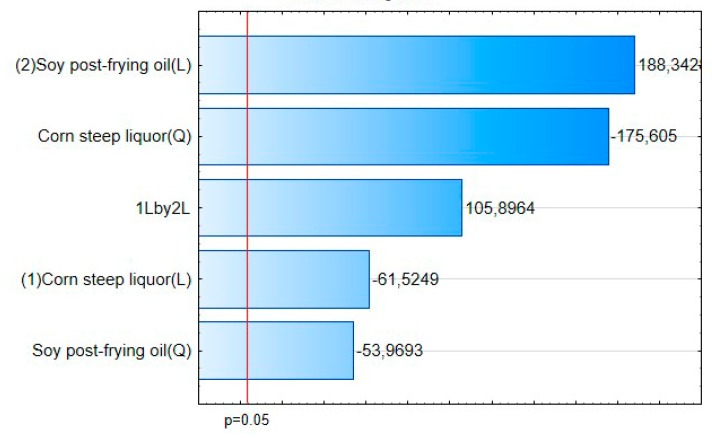
Influence of the independent variables (corn steep liquor and frying residual oil) and their interactions on engine burned oil EA.

**Table 1 biomolecules-10-00365-t001:** Levels of experimental design of the central rotational composite design (RCCD 2^2^).

Factors	Levels				
	−1.41	−1	0	+1	+1.41
Corn steep liquor	3.17	4	6	8	8.82
Frying waste oil	0.58	1.0	2	3	3.41

**Table 2 biomolecules-10-00365-t002:** RCCD 2^2^ factorial design containing corn steep liquor and waste cooking oil as independent variables and results for variables responses on emulsification activity and emulsification index.

Assays	Components Culture Medium		Emulsification Index (EI _24_ )				Emulsification Activity (UE)	
	Waste Cooking Oil	Corn Steep Liquor	Canola Oil	Corn Oil	Burned Engine Oil	Petroleum	Burned Engine Oil	Canola Oil
1	+1(3)	+1(8)	60.0	60.0	80.0	100.0	2.830	1.508
2	+1(3)	−1(4)	100.0	80.0	90.0	100.0	2.263	1.106
3	−1(1)	+1(8)	50.0	60.0	90.0	100.0	2.308	1.305
4	−1(1)	−1(4)	70.0	50.0	90.0	100.0	2.273	1.224
5	0(2)	+1.41(8.82)	60.0	60.0	80.0	100.0	2.802	1.401
6	+1.41(3.41)	0(6)	60.0	60.0	80.0	100.0	1.963	1.264
7	0(2)	−1.41(3.17)	90.0	50.0	90.0	100.0	2.279	1.437
8	−1.41(0.58)	0(6)	70.0	60.0	80.0	100.0	2.634	1.377
9	0(2)	0(6)	80.0	60.0	90.0	100.0	2.644	1.648
10	0(2)	0(6)	80.0	60.0	90.0	100.0	2.648	1.654
11	0(2)	0(6)	70.0	70.0	80.0	100.0	2.650	1.645
12	0(2)	0(6)	70.0	60.0	80.0	100.0	2.648	1.649

**Table 3 biomolecules-10-00365-t003:** ANOVA results for the quadratic model of AE in canola oil.

ANOVA; Var.: Canola oil (AE); R-sqr = 0.98329; Adj: 0.96936 (2**(2) central composite, nc = 4 ns = 4 n0 = 2 factors, 1 Blocks, 12 Runs; MS Pure Error = 0.000851 DV: Canola oil (AE)					
Factor	SS	df	MS	F	*p*
(1) Soy post-frying oil (L)	0.00096	1	0.000096	0.1122	0.754446
Soy post-frying oil (Q)	0.077905	1	0.077905	91.5448	0.000667
(2) Corn steep liquor (L)	0.017808	1	0.017808	20.9254	0.010226
Corn steep liquor (Q)	0.160956	1	0.160956	188.7146	0.000163
1L by 2L	0.093660	1	0.093660	110.0591	0.000467
Lack of fit	0.003143	2	0.001571	1.8464	0.230371
Pure error	0.003404	4	0.000851		
Total SS	0.391656	11			

SS = sum of square; df = degrees of freedom; MS = mean square; F = F test; *p* = value.

**Table 4 biomolecules-10-00365-t004:** ANOVA results for the quadratic model of AE in burned engine oil.

ANOVA; Var.: Burned engine oil; R-sqr = 0.65802; Adj: 0.37303 (DCCR NT 2) 2 factors, 1 Blocks, 12 Runs; MS Pure Error = 0.0000063 DV: Burned engine oil					
Factor	SS	df	MS	F	*p*
(1) Corn steep liquor (L)	0.023974	1	0.023974	3785.31	0.000009
Corn steep liquor (Q)	0.195301	1	0.195301	30836.94	0.000000
(2) Soy post-frying oil (L)	0.224662	1	0.224662	35473.00	0.000000
Soy post-frying oil (Q)	0.018447	1	0.018447	2912.69	0.000014
1L by 2L	0.071022	1	0.071022	11214.04	0.000002
Lack of fit	0.268831	3	0.089610	14148.98	0.000001
Pure error	0.000019	3	0.000006		
Total SS	0.786152	11			

SS = sum of square; df= degrees of freedom; MS = mean square; F = F test; *p* = value.

**Table 5 biomolecules-10-00365-t005:** Emulsification of motor oil and canola oil stability for 120 days on acid, neutral, and alkaline pH conditions (data expressed as mean ± standard deviation).

Time (Days)	Emulsification Index (EI_24_)					
	pH 4		pH 7		pH 9	
	Motor Oil	Canola Oil	Motor Oil	Canola Oil	Motor Oil	Canola Oil
1	98.3 ± 1.2	99.0 ± 0.7	91.0 ± 0.4	80.2 ± 0.2	90.7 ± 0.2	70.3 ± 0.2
15	96.3 ± 1.1	98.1 ± 1.3	91.0 ± 0.4	80.8 ± 0.2	85.4 ± 0.1	60.5 ± 0.4
30	85.1 ± 0.4	70.5 ± 0.4	85.4 ± 0.2	80.8 ± 0.6	80.4 ± 0.2	60.5 ± 0.4
60	80.2 ± 0.1	70.5 ± 0.0	85.4 ± 0.2	80.5 ± 0.0	80.3 ± 0.2	60.8 ± 0.3
90	80.7 ± 0.5	70.4 ± 0.5	85.3 ± 0.2	80.1 ± 0.1	80.2 ± 0.1	60.6 ± 0.5
120	80.0 ± 0.4	70.5 ± 0.7	85.3 ± 0.2	80.2 ± 0.1	80.2 ± 0.1	60.5 ± 0.3

**Table 6 biomolecules-10-00365-t006:** Motor oil and canola oil emulsification stability over 120 days on different salinity concentrations (data expressed as mean ± standard deviation).

Time (Days)	Emulsification Index (I_24_)					
	2% NaCl		15% NaCl		30% NaCl	
	Motor Oil	Canola Oil	Motor Oil	Canola Oil	Motor Oil	Canola Oil
1	98.4 ± 0.2	97.4 ± 0.4	70.2 ± 0.2	75.2 ± 0.2	60.4 ± 0.1	70.1 ± 0.0
15	95.1 ± 0.0	95.1 ± 0.1	60.3 ± 0.1	70.2 ± 0.2	50.6 ± 0.2	70.5 ± 0.0
30	90.2 ± 0.2	90.6 ± 0.2	62.1 ± 0.1	70.1 ± 0.2	50.6 ± 0.1	70.5 ± 0.0
60	90.2 ± 0.2	91.1 ± 0.1	60.3 ± 0.2	60.4 ± 0.2	57.3 ± 4.5	60.6 ± 0.4
90	90.5 ± 0.4	92.4 ± 0.4	60.2 ± 0.2	65.2 ± 0.1	50.5 ± 0.3	60.5 ± 0.0
120	90.2 ± 0.2	92.0 ± 1.3	61.4 ± 0.1	65.2 ± 0.1	50.4 ± 0.4	60.6 ± 0.2

**Table 7 biomolecules-10-00365-t007:** Motor oil and canola oil emulsification stability for 120 days on different temperatures treatments (data expressed as mean ± standard deviation).

Time (Days)	Emulsification Index (EI_24_)					
	4 °C		60 °C		121 °C	
	Motor Oil	Canola Oil	Motor Oil	Canola Oil	Motor Oil	Canola Oil
0	95.3 ± 0.2	80.4 ± 0.2	98.3 ± 0.2	81.0 ± 0.4	95.2 ± 0.1	60.4 ± 0.2
15	94.4 ± 0.2	80.8 ± 0.8	90.5 ± 0.3	80.8 ± 0.2	95.6 ± 0.2	60.5 ± 0.0
30	90.2 ± 0.2	61.0 ± 0.1	91.7 ± 0.6	70.2 ± 0.2	70.2 ± 0.0	65.0 ± 0.1
60	71.0 ± 0.0	61.6 ± 0.2	89.5 ± 0.0	70.1 ± 0.0	70.7 ± 0.4	60.6 ± 0.2
90	70.2 ± 0.2	60.2 ± 0.1	90.4 ± 0.2	70.7 ± 0.3	70.4 ± 0.2	55.5 ± 0.3
120	70.1 ± 0.0	60.2 ± 0.1	90.2 ± 0.1	71.3 ± 0.3	70.8 ± 0.3	50.6 ± 0.4

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
