# Peer review of "Eco-Friendly Bioemulsifier Production by Mucor circinelloides UCP0001 Isolated from Mangrove Sediments Using Renewable Substrates for Environmental Applications"

_biomolecules, 2020, doi:10.3390/biom10030365_

Round 1

Reviewer 1 Report

The manuscript about "Eco-friendly bioemulsifier production by Mucor circinelloides UCP0001 isolated from mangrove sediments using renewable substrates for environmental applications" contains some important information for the method and material developers. The manuscript in the current form needs a minor revision.

The referee's opinion is that the texts should be corrected where those contain the 'Pareto diagram/chart'-like expression. Per definitionem, a Pareto-chart is a chart with two different kinds of diagrams. Although the Wikipedia usually is not the most sophisticated source of information but in this case, it is: "A Pareto or sorted histogram chart contains both columns sorted in descending order and a line representing the cumulative total percentage." The sorted bar graph - apart from the fact that in the manuscript the diagrams are not histograms, only bar charts - is not a

Pareto chart/diagram even if the significance limit is drawn with a line. The SI contains not really SI, and it can be incorporated into the main text, after a little cut, as supposed to be originally intended. This clearly seen in the numbering of the figures. The meaning of ANOVE does not need an explanation, i.e. the length of the manuscript will not become too long by the insertion of two figs. and a short text.

Please, consider these during the finalization of the manuscript.

Author Response

RESPONSE TO COMMENTS ARE IN RED COLOR:

The referee's opinion is that the texts should be corrected where those contain the 'Pareto diagram/chart'-like expression. Per definition, a Pareto-chart is a chart with two different kinds of diagrams. Although the Wikipedia usually is not the most sophisticated source of information but in this case, it is: "A Pareto or sorted histogram chart contains both columns sorted in descending order and a line representing the cumulative total percentage." The sorted bar graph - apart from the fact that in the manuscript the diagrams are not histograms, only bar charts - is not a Pareto chart/diagram even if the significance limit is drawn with a line.

Done. The changes relative to the expression “Pareto diagram” used wrongly were taken out of the paper.

The SI contains not really SI, and it can be incorporated into the main text, after a little cut, as supposed to be originally intended.

Done. The changes were made along with the proper abbreviations.

The meaning of ANOVE does not need an explanation, i.e. the length of the manuscript will not become too long by the insertion of two figs. and a short text.

Done. The information regarding ANOVE, both for the emulsification activity in canola oil and burnt engine oil, were transferred from the supplementary material to the manuscript, as well as their respective tables.

Reviewer 2 Report

The manuscript described a method of preparing a bioemulsifier agent from low-cost material and characterization of its properties. Unfortunately, the authors did not show a clear understanding of the emulsifier product they intended to synthesize although previous studies of the product(s) exist, nor did they seem to be capable to explain thoroughly their experimental results. 

Please find more specific comments below.

The written English language needs significant improvement. There are many grammar mistakes. Additionally, the content in the introduction was not presented in a logical manner.

Although it may not be essential, the planning of optimizing the bioemulsifier production (2.13) should be placed before the section 2.3, since section 2.3~2.12 described characterization methods that apply to products from each production(level) tested.

The authors used the images in Figure 1 to support the statement about the stability of the emulsions 24 hours after initial emulsification (Line 190-191). Images of initial emulsifications should be given as well for comparison.

Regarding the toxicity evaluation,  why did the authors test 3 low concentrations 0.1%, 0.% and 1% of the bioemulsifier? Are these low concentrations comparable to the concentrations to be used for environmental applications?

The interpretation of FT-IR data is incorrect. The peak at 3232 cm-1 is clearly due to the O-H stretch of the hydroxyl groups in carbohydrates, at least, not C-H stretch.  Also, it is inappropriate to state that there is “a bioemulsifier molecule” with a hydrophilic region and a hydrophobic tail.  Multiple molecules, or components, contribute to the overall emulsifying property of the product.

The authors presented response surface graphs but did not explain how those leaner, quadratic effects and/or interaction factors determined can guide the optimization of product properties (i.g. emulsification activity) for their intended applications.

In the Discussion section, there is a large amount of text reviewing previous studies with being connected to the results of the present research (line 305 to 340). To some degree, the content here overlaps with the introduction.

For the environmental applications as included in the title, a proof of concept should be provided. For example, a home-made hydrocarbon-contaminated soil sample can be used to test the effectiveness of the bioemulsifier prepared.

Author Response

RESPONSE TO COMMENTS IN BLUE COLOR:

1) The written English language needs significant improvement. There are many grammar mistakes. Additionally, the content in the introduction was not presented in a logical manner.

Adjusted in all manuscript.

2) Although it may not be essential, the planning of optimizing the bioemulsifier production (2.13) should be placed before the section 2.3, since section 2.3~2.12 described characterization methods that apply to products from each production(level) tested.

Adjusted.

- In Line 172- 176, (Materials and Methods), the Table 1 was removed and inserted in line 103- 104 (item 2.3 Bioemulsifier Production) of the manuscript.

3) The authors used the images in Figure 1 to support the statement about the stability of the emulsions 24 hours after initial emulsification (Line 190-191). Images of initial emulsifications should be given as well for comparison.

Sorry!  The literature recommends as the minimum time to measure emulsion stability 24 hours (Cooper and Goldenberg 1987; Cirigliano and Carman, 1985; Santos et al., 2014; Souza et al., 2016; Pele et al., 2019; Rahman, et al., 2019). Therefore, the measurement performed in this study (24h after emulsion formed) is equivalent to the initial image of the emulsion.

References:

- Cooper, D.G.; Goldenberg, B.G. Surface-active agents from two Bacillus species. Appl Environ Microbiol. 1987, 53, 224-229. doi: 0099-2240/87/020224-06$02.00/0.

- Cirigliano, M.C.; Carman, G.M. Isolation of a bioemulsifier from Candida lipolytica. Appl Environ Microbiol. 1985, 48, 747-750. doi: 0099-2240/84/100747-04$02.00/0

-Souza, A.F.; Rodriguez, D.D.; Rubio-Ribeaux, D.; Luna, M.A.C.; Lima e Silva, T.A.; Andrade, R.F.S.; Gusmão, N.B.; Campos-Takaki, G.M. Waste Soybean Oil and Corn Steep Liquor as Economic Substrates for Bioemulsifier and Biodiesel Production by Candida lipolytica UCP 0998. Int. J. Mol. Sci. 2016, 17, 1608. doi: 10.3390/ijms17101608.

- Santos, D.K.F.; Brandão, Y.B.; Rufino, R.D.R.; Luna, J.M.; Salgueiro, A.A.; Santos, V.A.; Sarubbo, L.A. Optimization of cultural conditions for biosurfactant production from Candida lipolytica. Biocatal Agric Biotechnol. 2014, 3, 48-57. doi: 10.1016/j.bcab.2014.02.004

- Pele, M. A., Ribeaux, D. R., Vieira, E. R., Souza, A. F., Luna, M. A., Rodríguez, D. M., ... & Santiago, A. L. Conversion of renewable substrates for biosurfactant production by Rhizopus arrhizus UCP 1607 and enhancing the removal of diesel oil from marine soil. Electron j biotechn 2019;38:40-48. DOI: https://doi.org/10.1016/j.ejbt.2018.12.003

- Rahman, P. K., Mayat, A., Harvey, J. G. H., Randhawa, K. S., Relph, L. E., & Armstrong, M. C. Biosurfactants and Bioemulsifiers from Marine Algae. In The Role of Microalgae in Wastewater Treatment 2019, p. 169-188. Springer, Singapore. DOI:  https://doi.org/10.1007/978-981-13-1586-2_13

4) Regarding the toxicity evaluation, why did the authors test 3 low concentrations 0.1%, 0.5% and 1% of the bioemulsifier? Are these low concentrations comparable to the concentrations to be used for environmental applications?

In preliminary test of toxicity evaluation of secondary metabolites as bioemulsifier, the literature recommends the use of minimal concentrations for initial studies (Pelletier, et al., 2004; Camacho-Chab et al., 2013). In addition, the toxicity test in this study was evaluated by critical micellar concentration (CMC) of bioemulsifier that was 1%, as well as the half of it (0.5%) and a value ten times below it (0.1%).

References:

- Pelletier E., Delille D., Delille B. Crude oil bioremediation in sub-Antarctic intertidal sediments: Chemistry and toxicity of oiled residues. Mar. Environ. Res. 2004;57:311–327.

- Camacho-Chab JC, Guézennec J, Chan-Bacab MJ, et al. Emulsifying activity and stability of a non-toxic bioemulsifier synthesized by Microbacterium sp. MC3B-10. Int J Mol Sci. 2013;14(9):18959–18972. Published 2013 Sep 13. doi:10.3390/ijms140918959

5) The interpretation of FT-IR data is incorrect. The peak at 3232 cm-1 is clearly due to the O-H stretch of the hydroxyl groups in carbohydrates, at least, not C-H stretch.  Also, it is inappropriate to state that there is “a bioemulsifier molecule” with a hydrophilic region and a hydrophobic tail.  Multiple molecules, or components, contribute to the overall emulsifying property of the product.

Adjusted!

- In line 299- 303, (Results) the sentence " In addition, peak 3232 exhibited C-H stretch vibrations of fatty acid CH3 and =CH2 functional groups. These results demonstrate the hydrophobic tail composition and the molecule showed carbohydrate–protein–lipid complexes. The confirmation of carbohydrate, protein and fatty acid components in the isolated bioemulsifier confers upon them better emulsifying potential and ability to stabilize emulsions” Was Replaced By Sentence “In addition, peak 3232 exhibited O-H stretch of the hydroxyl groups in carbohydrates. The confirmation of carbohydrate, protein and fatty acid components in the isolated bioemulsifier confers upon them better emulsifying potential and ability to stabilize emulsions.”

6) The authors presented response surface graphs but did not explain how those leaner, quadratic effects and/or interaction factors determined can guide the optimization of product properties (i.g. emulsification activity) for their intended applications.

Adjusted!

Was added on line 222:

In addition, by detecting that the quadratic effects for the two independent variables are considerable, this indicates that the optimal levels are within the range of the experimental region.

Similar to the result observed for the emulsification of canola oil, the high significant values of the quadratic effect reinforced that the optimal levels were observed in the central points.

7) In the Discussion section, there is a large amount of text reviewing previous studies with being connected to the results of the present research (line 305 to 340). To some degree, the content here overlaps with the introduction.

Adjusted! Line 318-321.

8) For the environmental applications as included in the title, a proof of concept should be provided. For example, a home-made hydrocarbon-contaminated soil sample can be used to test the effectiveness of the bioemulsifier prepared.

Sorry! The term 'environmental applications "used in this study is linked the possible application of the bioemulsifier of  Mucor circinelloides by absence of toxicity  after test using the  Chlorella vulgaris and  Artemia salina. Based in this analysis, is possible indicate the promising potential of environmental application of the bioemulsifier produced by  Mucor for being these rganisms (Clorella vulgaris and Artemia salina) native of aquatic environments susceptible the contamination.

Reviewer 3 Report

The novelty of the study is not clearly explained. Aim of this work at the end of introduction is unclear and must be correlated with the novelty of this study. Please provide a clear objective/hypothesis in Introduction Section! In general, a clear thread should be woven through the text linking the introduction, methods, results, and conclusions. The Introduction could probably be a bit clearer with more references.
3. After carefully checking the article, I would suggest the following improvements of the article:
- First at all, some phrases/sentences/ are not clear, some words are not properly used, and chemical formula are incorrect written.

E.g.

Line 95 – the chemical formula of KH2PO4 and MgSO4.7H2O must be replaced with KH2PO4 and MgSO4·7H2O;

Line 96 – in the sentence “………………proposed by [7]…..” must be replaced with “…….proposed by Luna et al [7]…”; the same observation is on the line 124 “ …. as proposed by [11]”; and more mistakes in the whole article! please check and replace with the correct sentences!

Line 160 – The subsection “2.12 Characterization and determination of the Ionic Charge” – please reformulated!

Line 161 – In the sentence “The functional groups of the bioemulsifier were identified by the infrared spectroscopy technique” please remove technique;

Line 162 - In the sentence ” Infrared spectra were recorded on a Mattson 1000 FT-England FT-IR system..” please change the word "system" with spectrometer;

Line 191 – The sentence “Figure 1-B shows the optical microscopy analysis of the 191 emulsion formed between the burnt engine oil and the M. circinelloides metabolite was performed” must be reformulated;

Line 195 – The sentence “Also noteworthy in the microscopic image is the formation of the physical stability of the emulsion provided by the aggregation of dispersed particles, in contrast, without the presence of coalescence phenomenon was observed” must be reformulated and explained;

……………and so on!

Lines 294 -303 the FTIR investigation of bioemulsifier must be rewritten with attention because are more mistakes and problems of expression! Be careful! In its current state, these errors distract from the science within.

From my point of view the discussions must be improved and rearranged being correlated with experimental part. To increase the scientific level of this article for publication, I recommend relevant discussions in this regard.

-       Conclusions section - please improve this section with novelty of this study!

-       Overall, I feel that the entire submission requires further editing for content and language quality.

Author Response

REVIEWER: 3

RESPONSE  TO  COMMENTS IN ORANGE COLOR:

Line 95 – the chemical formula of KH2PO4 and MgSO4.7H2O must be replaced with KH2PO4and MgSO4·7H2O;

            Done. Line 96 KH2PO4 (0.02%), MgSO4.7H2O (0.02%).

Line 96 – in the sentence “………………proposed by [7]…..” must be replaced with “…….proposed by Luna et al [7]…”; the same observation is on the line 124 “ …. as proposed by [11]”; and more mistakes in the whole article! please check and replace with the correct sentences!

Done.

Line 160 – The subsection “2.12 Characterization and determination of the Ionic Charge” – please reformulated!

Done. Line 161: “Characterization and determination of the Ionic Charge”

It was replaced by “Characterization and ionic character”.

Line 161 – In the sentence “The functional groups of the bioemulsifier were identified by the infrared spectroscopy technique” please remove technique;

Done. Line 162: “The functional groups of the bioemulsifier were identified by the infrared spectroscopy technique”

It was replaced by “The functional groups of the bioemulsifier were identified by the infrared spectroscopy”.

Line 162 - In the sentence ” Infrared spectra were recorded on a Mattson 1000 FT-England FT-IR system..” please change the word "system" with spectrometer;

Done. Line 163: “Infrared spectra were recorded on a Mattson 1000 FT-England FT-IR system”

It was replaced by “Infrared spectra were recorded on a Mattson 1000 FT-England FT-IR spectrometer within the wavelength range of 500-4000 cm-1.”

Line 191 – The sentence “Figure 1-B shows the optical microscopy analysis of the 191 emulsion formed between the burnt engine oil and the circinelloides metabolite was performed” must be reformulated;

Done. Line 191: “Figure 1-B shows the optical microscopy analysis of the 191 emulsion formed between the burnt engine oil and the M. circinelloides metabolite was performed”

It was replaced by “Figure 1-B shows the optical microscopy analysis of the emulsion formed between the burnt engine oil and the M. circinelloides metabolite.”

Line 195 – The sentence “Also noteworthy in the microscopic image is the formation of the physical stability of the emulsion provided by the aggregation of dispersed particles, in contrast, without the presence of coalescence phenomenon was observed” must be reformulated and explained;

Done. Line 195: “Also noteworthy in the microscopic image is the formation of the physical stability of the emulsion provided by the aggregation of dispersed particles, in contrast, without the presence of coalescence phenomenon was observed”

It was replaced by “Also, the formation of the physical stability of the emulsion provided by the aggregation of dispersed particles is remarkable in the microscopic image, however, the absence of coalescence phenomenon was observed.”

Lines 294 -303 the FTIR investigation of bioemulsifier must be rewritten with attention because are more mistakes and problems of expression! Be careful! In its current state, these errors distract from the science within.

Done. Line 309: On the other hand, the peak 1652 cm-1 refers to the characteristic amide band of α or β secondary structures of proteins. These results prove the constitution of the functional groups belonging to the hydrophilic region of the bioemulsifier molecule. In addition, peak 3232 exhibited O-H stretch of the hydroxyl groups in carbohydrates. The confirmation of carbohydrate, protein and fatty acid components in the isolated bioemulsifier assigns them better emulsifying potential and ability to stabilize emulsions.

Conclusions section - please improve this section with novelty of this study!

The stability and non-toxicity of M. circinelloides UCP0001 bioemulsifier are decisive factors for its implementation in biotechnological applications in agriculture and bioremediation since this molecule has great potential for green technology, carefully designed by methods as a novel microbial emulsifier likely to instigate new collaborative approaches for protection of the biodiversity.

Round 2

Reviewer 2 Report

The authors did address some of my comments from the first round of review. However, they still need to work on their rational presentation of the content in the manuscript. In addition, the manuscript still needs professional English editing to correct grammar mistakes through the main text ( to list a few, line 46, line 85-87,line 115, line 126-127 and so on). There are multiple one-sentence paragraphs in the text, which also can be improved with professional editing.

The content in the introduction was still not presented in a logical manner. For example, the paragraph (line 53-57) should be presented right after the definition of the emulsifiers and surfactants (line 48, right after [2]), before introducing “bioemulsifiers” Previously I suggested placing section 2.13 before section 2.3, the authors moved Table 1 from section 2.13 to the end of section 2.3 but did not relocate the text that describes the experimental design along with the table. Reasonably, 2.13 and table 1 shall follow section 2.2 immediately, right before section 2.3. Line 97, the author's name should be given for reference [7]. Line 124, “…using ethanol(v/v 1:2),” what is the other solvent mixed with ethanol? Line 316, the wavenumber 3232 should have the unit cm-1 In the Discussion section, the authors need to work on the transitions between paragraphs. Currently, the flow of content is neither smooth nor showing the logic of their research.

Author Response

RESPONSE  TO REFERE 1

The manuscript about "Eco-friendly bioemulsifier production by Mucor circinelloides UCP0001 isolated from mangrove sediments using renewable substrates for environmental applications" contains some important information for the method and material developers. The manuscript in the current form needs a minor revision.

The referee's opinion is that the texts should be corrected where those contain the Pareto grphic/chart'-like expression. Per definitionem, a Pareto-chart is a chart with two different kinds of diagrams. Although the Wikipedia usually is not the most sophisticated source of information but in this case, it is: "A Pareto or sorted histogram chart contains both columns sorted in descending order and a line representing the cumulative total percentage." The sorted bar graph - apart from the fact that in the manuscript the diagrams are not histograms, only bar charts - is not a

We revised.

Pareto chart/diagram even if the significance limit is drawn with a line. The SI contains not really SI, and it can be incorporated into the main text, after a little cut, as supposed to be originally intended. This clearly seen in the numbering of the figures. The meaning of ANOVE does not need an explanation, i.e. the length of the manuscript will not become too long by the insertion of two figs. and a short text.

We revised.

Please, consider these during the finalization of the manuscript.

Reviewer 3 Report

The paper was improved and I recommend the article publication.

Author Response

RESPONSE  TO REFEREE 2

The authors did address some of my comments from the first round of review. However, they still need to work on their rational presentation of the content in the manuscript. In addition, the manuscript still needs professional English editing to correct grammar mistakes through the main text ( to list a few, line 46, line 85-87,line 115, line 126-127 and so on). There are multiple one-sentence paragraphs in the text, which also can be improved with professional editing.

We revised.

The content in the introduction was still not presented in a logical manner. For example, the paragraph (line 53-57) should be presented right after the definition of the emulsifiers and surfactants (line 48, right after [2]), before introducing “bioemulsifiers” Previously I suggested placing section 2.13 before section 2.3, the authors moved Table 1 from section 2.13 to the end of section 2.3 but did not relocate the text that describes the experimental design along with the table. Reasonably, 2.13 and table 1 shall follow section 2.2 immediately, right before section 2.3. Line 97, the author's name should be given for reference [7]. Line 124, “…using ethanol(v/v 1:2),” what is the other solvent mixed with ethanol? Line 316, the wavenumber 3232 should have the unit cm-1 In the Discussion section, the authors need to work on the transitions between paragraphs. Currently, the flow of content is neither smooth nor showing the logic of their research. 

We revised.
